# Towards Policy-aware World Models

## Abstract

World models have received significant attention from the robotics and computer vision community, both of whom have started scaling to networks comprising billions of parameters in the hope of unlocking new robot skills. In this paradigm, models are pre-trained on internet-scale data and then fine-tuned on robot data to learn policies. However, it is still unclear *what makes a good world model for downstream policy learning*. We show that world model prediction loss is in many instances uncorrelated with policy performance, forcing practitioners to train models to completion for correct evaluation. This results in slow, costly iterations of model training and policy evaluation. In this work, we demonstrate that the expected signal-to-noise ratio (ESNR) of policy gradients provides a reliable training-time metric for downstream policy performance. This provides a handle on the world model's *policy awareness*, which denotes how well a policy can learn from a model. We show that ESNR can be used to understand (1) when world models are sufficiently pre-trained, (2) how architecture changes affect downstream performance and (3) what is the best policy learning method for a given world model. Crucially, ESNR can be computed on-the-fly with minimal overhead and without a trained policy. We validate our metric on traditional architectures and tasks as well as large pretrained world models, demonstrating the practical utility of ESNR for practitioners who wish to train or finetune such models for robot applications. Visualizations and code available here: https://policy-aware.github.io/paper-anon.

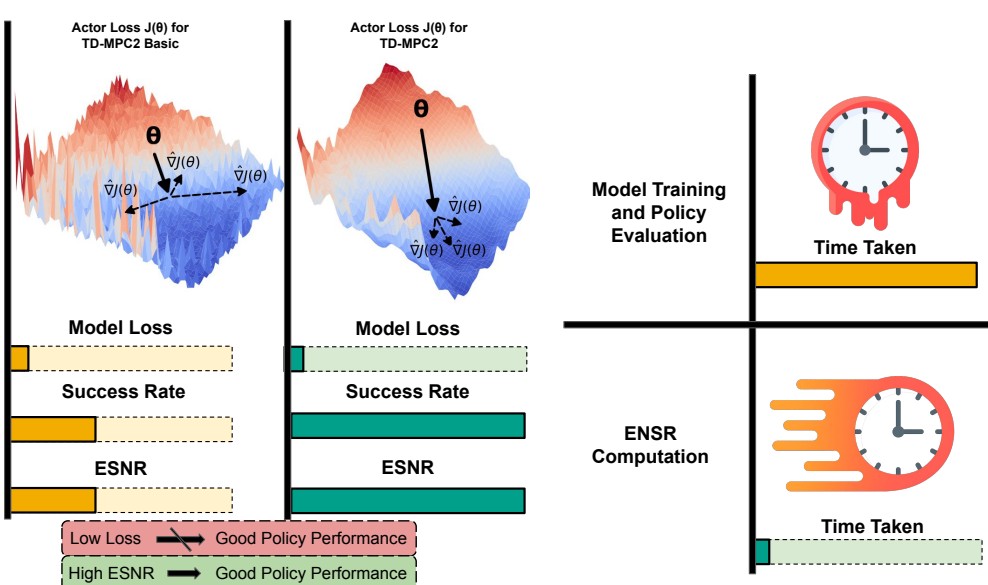

**Figure 1: Overview.** We find that different world model architectures can reach similar loss values under the same loss function, yet these values do not predict downstream policy performance. Instead, the Expected Signal-to-Noise Ratio (ESNR) of actor gradients $\nabla J(\theta)$ correlates strongly with final performance. ESNR can be computed quickly during training, orders of magnitude faster than full model training or policy evaluation, and captures the smoothness of the optimization landscape, providing an efficient surrogate for policy learnability.

# 1 INTRODUCTION

Large pretrained world models have delivered promising results for robot control (Assran et al. (2025), NVIDIA et al. (2025)). Presently, the most common training recipe involves pretraining a world model on a large corpus of unstructured data, which gives the model basic understanding of world dynamics, and then finetuning it on a smaller state action dataset to embed knowledge of how action influences state. With an increased interest in improving the efficacy of such models without expending valuable resources, many works have sought to understand what makes a better world model from the perspective of representation (Nair et al. (2022), Xiao et al. (2022), Assran et al. (2025))

However, the only way of evaluating the quality of such models with respect to downstream policy performance is to finetune it until convergence and then execute the policy in simulation or a real environment, often requiring days or weeks of training and evaluation. Despite being the de facto procedure in evaluating world models, this costly approach limits the rapid testing of alternative architectures and subsequent policy extraction methods. We use *policy extraction* to mean algorithms that utilize world model to drive executable control policy, e.g. online planning (CEM/MPPI) or policy-gradient methods (zeroth-/first-order).

Towards this end, we propose Expected Signal-to-Noise Ratio (ESNR) as a training-time metric to identify the *potential* downstream performance of world models. ESNR requires neither a trained policy nor environment rollouts, drastically reducing the wall-clock time and compute requirements relative to standard robot evaluation. In this paper, we highlight key empirical properties of ESNR: (1) **Training readiness**. ESNR behavior across training signals when the world model is ready for policy extraction. (2) **Architecture Ranking.** ESNR discriminates between world model architectures, providing a proxy ranking for expected policy performance. (3) **Policy Extraction Selection.** ESNR can guide the choice of policy extraction method for a given world model. This allows researchers to avoid excruciatingly long cycle times and rapidly iterate across different model architectures and converge to the best one.

Through this paper, we build concrete evidence of ESNR efficacy as a *downstream policy performance metric* on a variety of vision-based world model architectures, both traditional and recent SoTA large models. Additionally, we study the most common policy extraction methods, Zeroth-order policy gradients, First-order policy gradients, online planning, on a variety of continuous control tasks. Finally, we scale our experiments to **4** pre-trained world models representations – *ResNet, Dino, R3M* encoder world models (He et al. (2015), Nair et al. (2022), Zhou et al. (2025)) and *VJEPA2* (Assran et al. (2025)) – demonstrating ESNR's practical utility when designing large foundation world models *in the wild*.

# 2 RELATED WORK

World models for robotics have advanced rapidly, building on a long line of model-based control and representation learning. Early works like PILCO showed that learned dynamics can yield highly sample-effcient policy learning on real robots (Deisenroth & Rasmussen (2011), while PETS popularized probabilistic ensembles with MPC for robust control (Chua et al. (2018)). In parallel, the "World Models" framework of Ha & Schmidhuber (2018) leveraged the generative capabilties of such models and demonstrated the idea of learning policies "in imagination", leading to latent-dynamics works like PlaNet and Dreamer, which optimize policies entirely in latent space by predicting state reward/value and training the policy in an RL paradigm (Hafner et al. (2019), Hafner et al. (2020)). Complementarily, control-centric methods like TD-MPC combined learned latent dynamics models with online trajectory optimization at test time (Hansen et al. (2022)). The shared vision in the community is to learn a predictive model of the world that enables robots to plan or learn policies to solve complex tasks with far fewer real-world interactions.

Recent work scales this vision: Dino-WM combines a pretrained DinoV2 encoder with a forward dynamics predictor and performs online planning via a goal-reaching (Zhou et al. (2025)). Large foundation world models like VJEPA2 (Assran et al. (2025)) and Cosmos (NVIDIA et al. (2025)) scale the model and dataset size further to learn more general representations. Orthogonal to *how* models are used, a parallel line studies *what* representations help control: PVR vs. training-from-scratch comparisons Hansen et al. (2023b) and PVR within MBRL Schneider et al. (2025), along

with innovations in general-purpose encoders such as MVP, V-JEPA, and R3M (Xiao et al. (2022), Bardes et al. (2024), R3M (Nair et al. (2022)). Despite this progress, the model properties that actually enable downstream policy learning, beyond raw prediction quality, remains underexplored, motivating our focus on *policy awareness*.

There have been hints of works in this direction. Zhang et al. (2023) analyze policy gradients and link failure modes to exploding gradient variance arising from a lack of objective landscape regularization. SimbaV2 introduces regularization strategies to help improve policy performance in an RL setting, emphasizing the importance of regularization for policy gradient methods (Lee et al. (2025)). Parmas et al. (2023b) utilized the exptected signal-to-noise ratio (ESNR) to assess the quality of gradient estimators, and PWM used ESNR to show that regularized world models yield more reliable first-order policy gradients, guiding hyperparameter choices (Georgiev et al. (2025)). However, prior work has not established a single, practically useful metric for predicting downstream policy performance across heterogeneous world model architectures and policy extraction methods. In this work, we provide evidence that the policy-gradient ESNR of a pretrained world model can serve as such a metric. This focus on mechanistic properties that predict downstream success echoes broader trends in ML as a whole. For example, there exists a line of work relating generalization to loss-landscape geometry and shape (Keskar et al. (2017), Foret et al. (2021)), and scaling-law analyses that connect model and training-time properties to task performance (Kaplan et al. (2020), Hoffmann et al. (2022), Alabdulmohsin et al. (2022)). Our contribution brings a similar lens to world models for robotics: using ESNR to *anticipate* policy learnability before policy training.

## 3 BACKGROUND

### 3.1 WORLD MODELS

World models are a class of predictive models that aim to capture the underlying dynamics of an environment in a compact, structured representation. Rather than mapping observations state $s_t \in \mathcal{S}$ directly to actions $a_t \in \mathcal{A}$, world models learn latent states $z_t$ and an internal model of the environment transition dynamics. This model serves as a surrogate for the real environment, enabling an agent to simulate trajectories, reason counterfactually, and plan over imagined rollouts. We formalize the components below:

$$
\begin{array}{lll}
\text{Encoder} & z_t = E_\phi(s_t) & \triangleright \text{ Maps observation state to their latent representations} \\
\text{Latent dynamics} & z_{t+1} = F_\phi(z_t, a_t) & \triangleright \text{ Models (latent) forward dynamics} \\
\text{Decoder} & \hat{s}_{t+1} = D_\phi(s_{t+1}) & \triangleright \text{ Decodes latent } back \text{ to observation state}
\end{array}
\tag{1}
$$

Although all world models exhibit the same capabilities – predicting the next state from a history of states and action – they learn such inductive biases in different ways. A reconstruction-based world model learns an explicit generative model based on a reconstruction goal. Given a current observation and action, they minimize the *prediction loss* of the next observation:

$$
\mathcal{L}_{\text{rec}}(\phi, \psi, \theta) = \mathbb{E}_{(s_t, a_t, s_{t+1})_{0:H} \sim \mathcal{B}} \left[ \sum_{t=0}^{H} \ell\big(\hat{s}_{t+1}, s_{t+1} + \ell(...)\big) \right]
$$
$$
\text{where } \hat{s}_{t+1} = D(F(E(s_t), a)) \text{ is the reconstructed state at at time } t+1, \tag{2}
$$
$$
\ell \text{ is some difference function (e.g., MSE)},
$$
$$
\ell(...) \text{ is some task relevant loss (eg. reward prediction minimization)}
$$

A reconstruction-less world model learn an *implicit* generative model without the need for reconstruction. They instead guide the training with latent consistency and some task-relevant objective (eg. reward prediction, value prediction).

$$\mathcal{L}_{\text{no rec}}(\phi, \psi, \theta) = \mathbb{E}_{(s_t, a_t, s_{t+1})_{0:H} \sim \mathcal{B}} \left[ \sum_{t=0}^{H} \ell(\hat{z}_{t+1}, E(s_{t+1})) + \ell(...) \right]$$

where $\hat{z}_{t+1} = D(F(E(s_t), a))$ is the latent state at time $t+1$, (3)

$r$ is the ground truth environment reward

$\ell$ is some difference function (e.g., MSE),

$\ell(...)$ is some task relevant loss (eg. reward prediction minimization)

### 3.2 POLICY EXTRACTION FROM WORLD MODELS

After training a world model, there exist various ways to *extract* policies from the latent state representation. We assume that the policy is a parameterized stochastic function from which we can sample actions given the current state: $a_t \sim \pi_\theta(\cdot|z_t)$. Given some objective $J(\theta)$ such as reward maximization

$$J(\theta) = \mathbb{E}_{a_t \sim \pi(\cdot|z_t)} \left[ \sum_{t=1}^{\infty} \gamma^t R(z_t, a_t) \right] \tag{4}$$

our goal is to find the set of parameters $\theta$ that maximize $J(\theta)$. The most common approach is to solve it via gradient descent as popularized in deep learning. However, computing this expectation analytically is intractable and usually approximated via Monte Carlo (MC) sampling where $\hat{\nabla}^{[*]} J(\theta)$ is a single MC sample of some gradient estimator (designated by placeholder *).

$$\bar{\nabla}^{[*]} J(\theta) = \frac{1}{N} \sum_{n=1}^{N} \hat{\nabla}^{[*]} J(\theta) \tag{5}$$

The research community has converged on two main gradient estimators: REINFORCE (Zeroth-Order Gradients) and pathwise gradients (First-Order Gradients) (Sutton et al. (1999), Heess et al. (2015)). Zeroth-Order Gradients (ZoG) are popular in the RL community because they do not require the environment to be differentiable.

$$\nabla^{[0]} J(\theta) = \mathbb{E}_{a_t \sim \pi_\theta(\cdot|z_t)} \left[ J(\theta) \nabla_\theta \log \pi_\theta(a_t|z_t) \right] \tag{6}$$

In contrast, First-Order Gradients (FoG) provide lower variance gradient estimates but require a differentiable objective.

$$\nabla^{[1]} J(\theta) = \mathbb{E}_{a_t \sim \pi_\theta(\cdot|z_t)} \left[ \nabla_\theta J(\theta) \right] \tag{7}$$

Finally, another popular way to optimizing Eq. 4 is Online Planning, which does not require a parameterized policy but can be computationally expensive. Two popular gradient-free approaches here have been Cross-Entropy Method (CEM) and Model Predictive Path Integral (MPPI). Although not cast as gradient descent, these methods iteratively update a proposal distribution over action sequences iteratively – e.g., CEM via elite-set moment updates – yielding gradient-step-like behavior. For example, with CEM:

$$a_{0:H} \leftarrow \frac{1}{|\mathcal{E}|} \sum_{n \in \mathcal{E}} a_{0:H}^{(n)}, \quad a_{0:H}^{(n)} \sim \mathcal{N}(\mu, \Sigma) \tag{8}$$

Here, $\mathcal{E}$ denotes the elite set of samples with the highest return and $a_{0:H}$ is the current action sequence. We can formalize this as a gradient update, where the gradient is with respect to the objective $J_{\text{act}}$ which is conditioned on the action sequence.

$$a_{0:H} \leftarrow a_{0:H} - \alpha \nabla_{a_{0:H}}^{[\text{MPPI}]} J_{\text{act}}(a_{0:H}) \tag{9}$$

We defer the reader to the respective works for more details on CEM and MPPI (Rubinstein & Kroese (2004), Williams et al. (2015)).

## 3.3 METHODS

We evaluate the proposed method with representative methods for each paradigm. We select DreamerV3 (reconstruction-based) (Hafner et al. (2020)), TD-MPC2 (reconstruction-less) (Hansen et al. (2023a)), and PWM (Georgiev et al. (2025)) (recontruction-less + First order Gradients), three proven algorithms in their respective training strategies. We also add one additional variant of TD-MPC2, substituting LayerNorm and ReLU activation with Mish (Misra (2020)) and SimNorm (Lavoie et al. (2022)), to study how decreasing levels of world-model regularization affect downstream policy performance and SNR.

These world models are designed for the RL setting, where policy extraction is based on the presence of rewards. We then scale up our experiments to large-scale world models with encoders pre-trained on large corpus of data: ResNet, R3M, DinoV2, VJEPA-2 world models ((Russakovsky et al. (2015)), Nair et al. (2022), Oquab et al. (2023), Assran et al. (2025)). We present an overview of all methods and their respective policy extraction method in Table 1.

**Table 1:** Overview of world models that we consider.

| Model / Encoder | $\pi$-extraction |
|---|---|
| **RL World Models** | |
| • TD-MPC2 | Online planning |
| • TD-MPC2 *basic* | Online planning |
| • DreamerV3 | Zeroth-order gradients |
| • PWM | First-order gradients |
| **Large Pretrained World Models** | |
| • ResNet18 | Online planning |
| • R3M | Online planning |
| • DINO | Online planning |
| • VJEPA2 | Online planning |

# 4 MEASURING POLICY AWARENESS IN WORLD MODELS

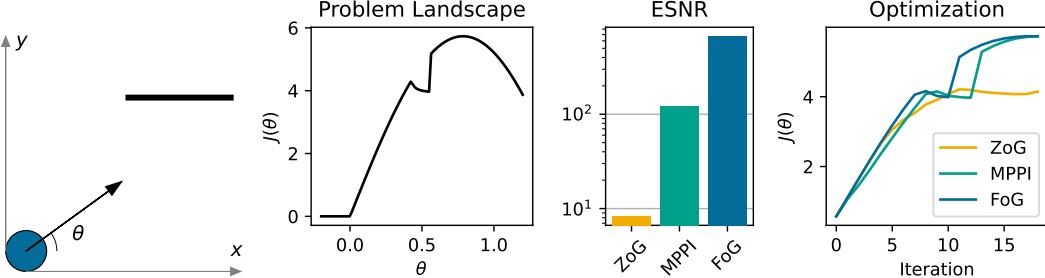

**Figure 2: Toy ESNR example.** In a ball-shooting task with a wall-induced discontinuity, ESNR over $\theta \in [-0.2, 1.2]$ ranks gradient estimators and predicts optimization speed: higher ESNR $\Rightarrow$ faster ascent.

For a world model to be useful for robot control, it must be (1) accurate and (2) induce a good optimization landscape for learning a policy. The former has been well studied in the video prediction model community where it is common to use PSNR, SSIM and LPIPS (Wang et al. (2004), Zhang et al. (2018)). In this work, we focus on the latter.

Parmas et al. (2023a) first introduced Expected Signal To Noise Ratio (ESNR) as a method of comparing different types of gradient estimators and overcome the issue that certain methods may utilize bias itself in estimating the policy gradients Hafner et al. (2020).

$$ESNR = \mathbb{E}_{o \sim \mathcal{O}} \left[ \frac{\mathbb{E}_{a \sim \pi_\theta(\cdot|o)}[\hat{\nabla}_\theta^{[*]} J(\theta)]^2}{\mathbb{V}_{a \sim \pi_\theta(\cdot|o)}[\hat{\nabla}_\theta^{[*]} J(\theta)]} \right] \quad (10)$$

If $\hat{g}_1$ and $\hat{g}_2$ are unbiased stochastic gradient estimators of $\nabla_\theta J(\theta)$ and $\text{Var}(\hat{g}_1) < \text{Var}(\hat{g}_2)$, then optimization with $\hat{g}_1$ is expected to converge faster. However, many modern methods such as DreamerV3 introduce gradient estimator bias via world-model surrogates as a learning signal for policy learning (Hafner et al. (2023)). ESNR overcomes this by instead computing the ratio between the signal (size of gradients) to noise (variance of gradients). In general, given two models have similar inductive bias, the one with a higher ESNR will produce *better* gradients. We build intuition of ESNR with a pedagogical task: a projectile is thrown forward with the goal of maximizing distance

traveled in the presence of a object, which introduces discontinuities in the problem landscape (Figure 2). We first compute the ESNR of different policy extractors over the full problem landscape. Starting from $\theta = 0$, we maximize $J(\theta)$ by gradient ascent and find that gradient estimators with higher ESNRs converge faster.

We propose using **ESNR as a test-time metric** to *predict* downstream policy performance. ESNR is computable during world model pretraining amd requires no trained policy or environment interaction. We believe this property is important to scaling up to world models with billions of parameters: an *a priori* metric enables rapid iteration across different world model architectures, allowing practitioners to identify high-performing designs more quickly. In Code 1 we provide a reference implementation that can be added to offline world model training. In this implementation, we compute the ESNR with respect to the action trajectory gradients rather than policy parameter gradients that Parmas et al. (2023a) utilized. This allows us to compare ESNR across parameteric policy extraction methods (FoG, ZoG) and non-parameteric policy extraction methods (online planning).

**Code 1:** ESNR pseudo-code where N is number of action samples and B is number of observation samples.

```python
def compute_esnr(actions, J, grad_f):
    """
    :param actions: tensor of shape (B, act_dim)
    :param J: function with signature J(actions) -> float
    :param grad_f: function with signature grad_f(J, actions) -> grads
    """
    actions = actions.detach().requires_grad_()
    grads = grad_f(J, actions)                    # (N, B, action_dim)
    grad_mean = grads.mean(dim=0)                 # (B, action_dim)
    grad_std = grads.std(dim=0)                   # (B, action_dim)
    snrs = grad_mean**2 / (grad_std**2 + 1e-8)    # (B, action_dim)
    return snrs.mean()
```

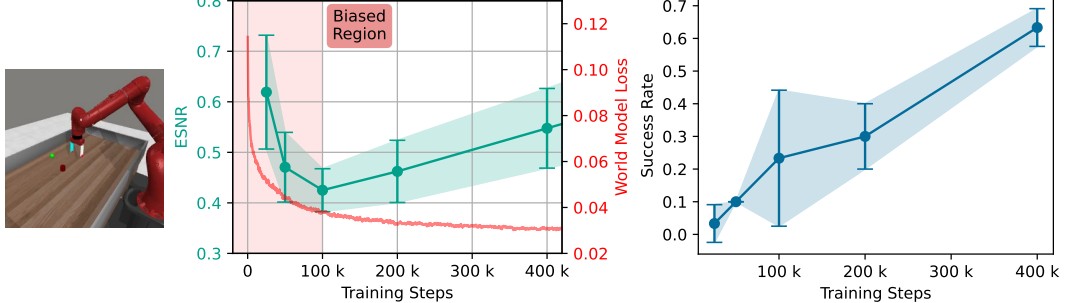

**Figure 3: ESNR over training.** TD-MPC2 offline on MetaWorld Push. **Middle:** policy ESNR (mean±sd, 10 seeds) during pretraining. **Right:** episodic success of policies extracted from checkpoints. World-model loss is not predictive; after 100k steps ESNR tightly correlates and serves as a surrogate, while for $<100$k a "biased region" persists.

Higher ESNR does not necessarily imply a better or sufficiently trained world model. For example, a trivial gradient estimator $\nabla^{[\infty]} J(\theta) = 0$ would have a misleading ESNR $\to \infty$. The case would be the same for a world model initialized with all $\theta = 0$. To build intuition of how ESNR behaves during world model training, we study it on MetaWorld Push task Yu et al. (2020) by applying TD-MPC( Hansen et al. (2023a)) to solve the task from camera observations. We first pre-train the world model on offline data and measure the ESNR over time. Figure 3 reveals a U-shape curve for ESNR which starts high, reduces to a minimum at 100k training steps and then grows until the end of training. We take each pre-trained checkpoint, learn a policy from it until convergence, and measure the success rate. We observe that ESNR becomes well correlated with success rate in the [100k, 400k] region. We refer to the training steps before the ESNR minimum as the *bias region* where ESNR is artificially high and the policy gradients are biased due to an inaccurate world model. As such, ESNR as a training time metric becomes useful only after it escapes this biased region.

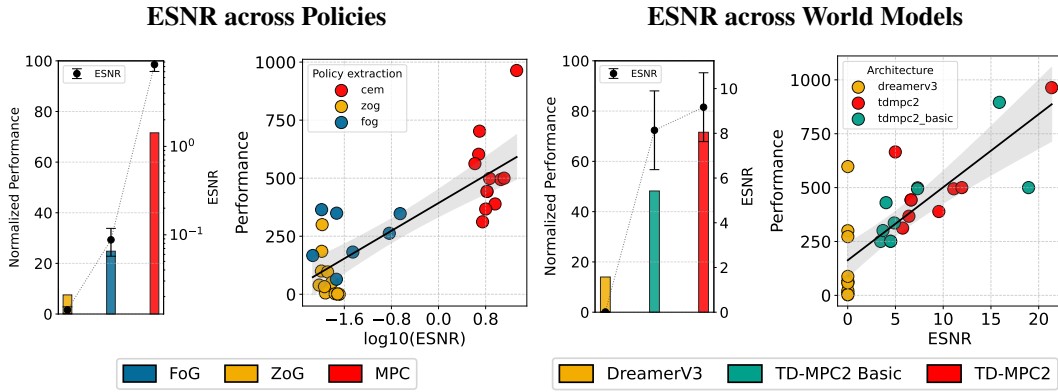

**Figure 4: ESNR and policy-extraction performance. Left:** bar and scatter plots show the ESNR–performance relationship across policy-extraction methods. **Right:** the same relationship across world-model architectures. ESNR error bars are 95% bootstrap CIs; bar-chart performance is normalized and averaged over all tasks.

## 5 EXPERIMENTS

In this section, we identify that prediction loss, which is prevalently used in the community to gauge whether a world model is ready for robot deployment, can be misleading. We posit that ESNR can serve as a viable alternative: we study ESNR's ability to (i) identify the best policy learning method (ii) rank world model architectures, and (iii) detect when a model is sufficiently pretrained. For all the experiments, we utilize the same ESNR definition outlined in Code 1. We evaluate the RL methods (refer to Table 1) on 12 tasks spanning 2 domains: DMControl

| Experiment | Spearman |
|---|---|
| ↔ $\pi$ methods | 0.7917 |
| ↔ WM archs | 0.8366 |

**Table 2: ESNR–performance correlations across axes.** Spearman correlations between ESNR and downstream performance when varying $\pi$-extraction methods and world model architectures.

(Tassa et al. (2018)) and Meta-World (Yu et al. (2020)). Combined, they comprise of a variety of object manipulation and locomotion tasks. To train our models, we use visual observations from both suites, and partial-propioceptive data in MetaWorld. Learning a policy is heavily dependent on imagined rollouts of the world model, which can qualitatively change based on the data distribution. Therefore, we use an offline training regime for all of our experiments: we first collect a fixed dataset for every task using TD-MPC2; we then pre-train *all* world models on this fixed dataset; lastly, we train policies or perform online planning exclusively with the pre-trained world model. This last stage can be termed *policy extraction*. Similar to recent large world-model such as Assran et al. (2025), the only interaction with the true environment is during evaluation which is subsequently how we assess performance on all experiments.

**Why is prediction loss not enough?**

Prediction loss alone can be misleading as a proxy for policy performance, especially when the world-model training objective is not aligned with policy learning. Our results in Figure 5 illustrate this: • TD-MPC2 *basic* attains lower prediction loss than • TD-MPC2, yet yields worse downstream performance on all MetaWorld tasks. • TD-MPC2 *basic* uses weaker regularization than • TD-MPC2 and consequently induces a sharper, more irregular policy objective landscape. We can parallel this observation to the ceiling-bounce toy problem in Figure 8, where faithfully modeling all discontinuities does not necessarily aid optimization. In both cases, prediction loss is agnostic to how learnable the resulting policy objective is, whereas ESNR correctly ranks the two world models in line with their policy performance. We further evaluate ESNR on larger world models (• DINO-WM and • VJEPA2) across 3 tabletop manipulation tasks (see further experiment details in Appendix C). A standard approach is to assess world-model quality using LPIPS between predicted and ground-truth representations. However, our results in Figure 5 show that LPIPS fails to reliably distinguish models that yield strong downstream policy performances, whereas ESNR identifies the world model that produces the best policy. We attribute this to ESNR being, by design, a policy-centric metric: it not only measures how well a model predicts its representations, but also how conducive those representations are to strong downstream policy performance.

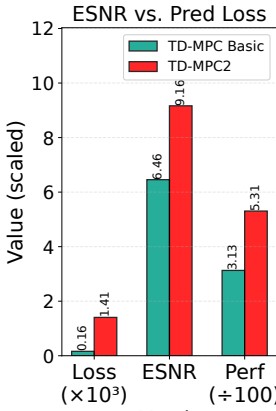
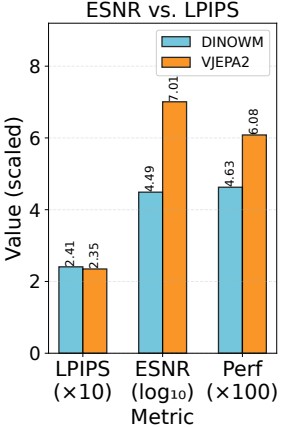

**Figure 5: ESNR vs. baselines and correlation summary. Left:** ESNR vs. prediction loss (left) and ESNR vs. LPIPS (right) across tasks. **Right:** Spearman Correlation of ESNR and standard model-quality metrics with downstream performance, split into *RL* (policy-learning tasks) and *Planning* (planning with a frozen world model).

### 1. Can ESNR be used to determine which policy extraction method is best?

Our results indicate that the ESNR strongly correlates with the episode returns achieved by different policy extraction methods. For this experiment, we fix a trained TD-MPC2 world model and evaluate three policy extraction paradigms: • MPC with Cross-Entropy Method • FoG and • ZoG. For each method, we extract the policy at a TD-MPC2 world model pretrained at 200K steps. Figure 4 and Table 2 show that ESNR strongly correlates with downstream performance. Further aggregating all task performances reveals a global correlation between ESNR and policy performance. FoG methods utilize ground truth gradients from a differentiable objective, so it is unsurprising that • FoG yields higher ESNR and returns than • ZoGs: the gradients will have lower variance and greater expected norm when the objective is well defined. The surprising result is that CEM consistently attains both the highest ESNR and policy performance. However, the fundamental tradeoff here is that learning-based approaches incur high upfront training cost but have low test-time cost, while online planning has low upfront cost but high test-time cost that scales with planning horizon and number of sampled trajectories.

### 2. Can ESNR be used to compare the downstream performance of world model architectures?

To assess ESNR's predictive power across different world model architectures, we train three world model architectures (• TD-MPC2 *basic* • TD-MPC2 • DreamerV3) for 200K gradient updates and evaluate policies as prescribed

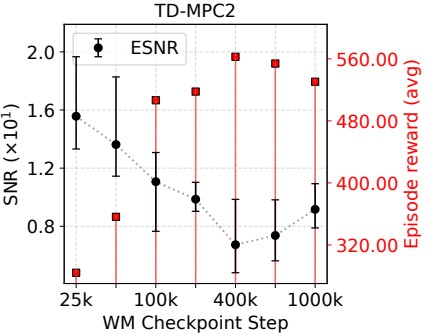

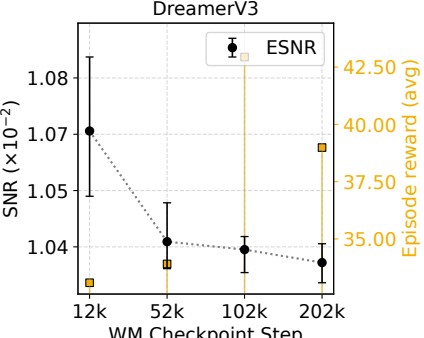

**Figure 6: ESNR vs. Policy Performance (TDMPC2, 20 ).** ESNR shows 95% bootstrap CIs; lollipops mark policy-extraction returns (3 seeds) at the corresponding WM checkpoint. Returns are averaged over selected MetaWorld tasks.

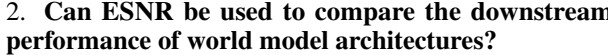

by each method. On average, ESNR aligns with the episode returns across architectures (see Figure 4, Table 2): • DreamerV3 underperforms on most tasks and shows the lowest ESNR. • TD-MPC2 architecture, which features stronger regularization compared to • TD-MPC2 *basic*, achieves higher returns and correspondibly higher ESNR. The world model substantially shapes the optimization landscape seen by the policy (Figure 1). Prior work linked objective landscape properties with RL performance (Lee & Yoon (2025), Xing et al. (2025)). We extend this understanding with a

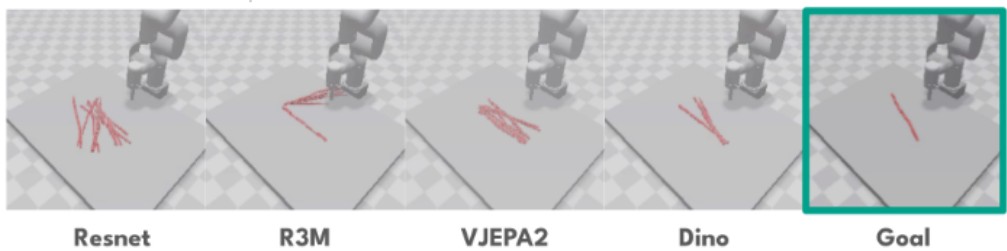

| Method | ESNR time (hrs) | Eval time (hrs) |
|---|---|---|
| ResNet | 0.003 | 0.70 |
| R3M | 0.003 | 2.21 |
| DinoWM | 0.027 | 0.72 |
| V-JEPA2 | 0.358 | 4.58 |

**Figure 7: ESNR & Performance of Large Pretrained Models. Left**: Performance $1/CD$ (lower CD is better) of different large pretrained models. **Right**: Wall time to evaluate ESNR vs. evaluation. **Bottom**: Qualitative comparisons on *rope* where we overlay observations from a full episode to track changes to the environment.

policy-centric lens: ESNR quantifies the effective signal avaliable to the policy under a given world model. Our experiments also underscore the importance of objective landscapes induced by world models: the performance gap between • TD-MPC2 and • TD-MPC2 *basic* (similar models but varying in regularization) suggests that the world model, not the policy learning method, is often the primary bottleneck.

3. **Key behaviors of ESNR across training.**

To further assess ESNR as a performance estimator, we track it over world model training steps. We saw in Section 4 that ESNR exhibits a U-shaped trend across epochs (decreasing early, then increasing afterwards). We scale up this observation to include all the chosen MetaWorld tasks and 2 methods (• TD-MPC2, • DreamerV3). For this experiment, we first pretrain the world model to convergence, save checkpoints at regular intervals, and, for each checkpoint, extract a policy across three seeds. The aim is to identify how ESNR and policy performance behave at different stages of training. In the earliest stages of world model training, the world model behaves almost like a random gradient generator, yielding large but unreliable gradient estimates. This is reflected by the artificially high ESNR in both models (see Figure 6). This is also supported by the fact that, at this stage of training, policy returns are low, contradicting the fundamental assumption that high ESNR implies higher returns. Through training, the ESNR gradually reduces until it reaches its minima. We also experience the highest returns at this stage of training. We hypothesize that at this point, our gradients are unbiased enough to support policy extraction. In practice, this suggests a simple procedure: during training, we monitor ESNR and trigger policy extraction once ESNR reaches its minimum and begins to increase, rather than waiting for a full sweep of checkpoints or relying solely on model loss.

4. **Does ESNR still serve as a reliable indicator in large vision-based world models with billions of parameters?**

As the field advances toward larger world models, the common consensus is that bigger model sizes are better. Although true in part, a caveat is that the inferencing these models remains exhaustively slow . VJEPA2 Assran et al. (2025) and Cosmos NVIDIA et al. (2025) underlined the drastically

slow action execution times required to perform online planning. Without good evaluation metrics, it will be difficult to scale and converge to the right architecture. We probe ESNR capabilities in predicting the performance of such large models. We finetune 4 pretrained world models (• ResNet • R3M • DINO-WM • VJEPA2) on an offline dataset comprising of observation-action transitions. We then perform online planning with all models using MPC with CEM optimization process. Each algorithm is allowed to step in the environment for 50 timesteps and we test across 10 different seeds. The performance is measured as the inverse of Chamfer Distance between the current state and goal state, where we take the lowest Chamfer Distance achieved in the entire environment rollout as our performance metric. Figure 7 indicates that the policy performance and ESNR are highly correlated. Furthermore, the table in Figure 7 highlights the demanding walltime requirements of inferencing and rolling out these models in simulator. On the other hand, we can calculate ESNR in a fraction of the time.

## 6 CONCLUSION

We introduced the Expected Signal-to-Noise Ratio (ESNR) of policy gradients as a training-time metric to anticipate the downstream policy performance of world models. Unlike traditional evaluation procedures that require fully training and deploying a policy, ESNR can be computed directly during world model training with minimal overhead. Across a wide range of experiments, we showed that ESNR provides actionable insights: it signals when a world model is sufficiently pretrained to support policy learning, distinguishes between model architectures with different inductive biases, and identifies which policy extraction method—zeroth-order, first-order, or online planning—is best suited for a given model. Extending our analysis to large pretrained vision-based world models, we found that ESNR remains a reliable proxy for policy quality even when standard evaluation is prohibitively expensive. Together, these results establish ESNR as a practical diagnostic for practitioners aiming to iterate quickly on world models and extract effective policies without incurring the full cost of downstream training and evaluation.

**Limitations and Future Work**:

While ESNR provides a powerful and efficient proxy for downstream policy performance, it must be used with care. ESNR is not meaningful when the underlying world model is fundamentally inaccurate, as biased gradients can yield deceptively high ESNR values. This limits its utility in the very early stages of training or when modeling assumptions are severely violated. Furthermore, we have not yet evaluated ESNR extensively on large-scale world model tasks with billions of parameters. Doing so remains challenging due to the significant computational demands of training and evaluating such models. Future work should explore applying ESNR in broader large-model settings, as well as investigating complementary metrics that account for model bias in addition to gradient variance.

## REPRODUCIBILITY STATEMENT

We are committed to ensuring the reproducibility of our results. To this end, we have open-sourced the full codebase used to conduct our experiments, as well as the datasets used for training. We also release all relevant pretrained model checkpoints to facilitate verification of our results and to enable further research. Detailed hyperparameter settings, compute resources, and random seeds are documented in the accompanying code release.

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

## A   TOY PROBLEM DETAILS

This section provides more details on the ceiling-bounce toy example used to showcase ESNR and its effects on stochastic optimization in Section 4. We constructed a simple problem of a point mass (ball) being launched from the origin $(0,0)$ with velocity $v$ at initial angle $\theta$. The goal is to maximize the horizontal distance traveled before the projectile lands on the ground ($y = 0$). The dynamics without contact are given by

$$x(t) = v\cos(\theta)t \qquad\qquad y(t) = v\sin(\theta)t + \tfrac{1}{2}gt^2,$$

where $g$ is the gravitational acceleration. A rigid horizontal ceiling is placed at height $y_s$ and spans the interval $[x_1, x_2]$. If the trajectory passes entirely below the ceiling, the ball follows standard ballistic motion and lands after

$$t_{\text{ground}} = \frac{-2v\sin(\theta)}{g},$$

yielding a final horizontal distance $J(\theta) = v\cos(\theta)\,t_{\text{ground}}$.

If the projectile intersects the ceiling before reaching the ground, a contact event occurs. The contact time $t_c$ is given by the smallest positive solution of

$$\tfrac{1}{2}gt_c^2 + v\sin(\theta)\,t_c - y_s = 0,$$

with the additional requirement that $x(t_c) \in [x_1, x_2]$ and $t_c < t_{\text{ground}}$. At impact, the vertical velocity is updated according to a restitution coefficient $e \in [0,1]$,

$$v_y^+ = -e\big(v\sin(\theta) + gt_c\big),$$

while the horizontal velocity remains unchanged. The ball then continues from $(x_c, y_s)$ with velocities $(v\cos(\theta), v_y^+)$ until it hits the ground. The post-contact flight time $\tau_g$ is obtained by solving

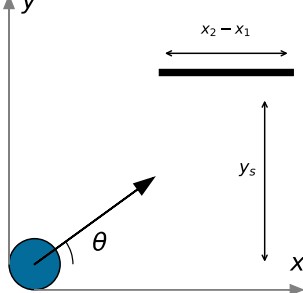

**Figure 8:** Pedagogical ceiling-bounce toy problem visualized.

$$y_s + v_y^+\tau_g + \tfrac{1}{2}g\tau_g^2 = 0,$$

and the total horizontal distance in the bounce case is

$$J(\theta) = x_c + v\cos(\theta)\,\tau_g.$$

This toy task produces a discontinuous objective landscape. Small variations in $\theta$ can switch contact on or off, leading to sharp changes in $J(\theta)$. This property makes the ceiling-bounce example a challenging benchmark for stochastic gradient estimators, which must handle both the variance and discontinuity of gradient signals. Problem parameters we used were $g = -9.81$, $x_1 = 1.0$, $x_2 = 2.5$, $y_s = 0.5$, $e = 0.5$.

For the ESNR estimation and optimization process, we used standard pytorch autograd with $N = 64$ samples and noise $\sigma = 0.03$ for the action sampling. We optimized the problem with each gradient estimator type for 20 iterations using learning rate $\alpha = 0.05$ and the Adam optimizer (Kingma & Ba, 2014).

## B   ESNR ALGORITHM

Across all experiments, we use a fixed budget of 500 environment seeds and 100 gradient samples per initial state. All remaining hyperparameters (e.g., horizon length, elite set size) are matched to each algorithm's original settings from the respective works.

---

**Algorithm 1 ESNR (Score / log-prob form)**: SNR of REINFORCE gradients induced by a world model

---

1: **Inputs:** world model $\mathcal{M}$ (encode, reward, next), horizon $H$, discount $\gamma$, MC samples $S$
2: **Policy (per step):** $a_t \sim \pi_\theta(\cdot \mid z_t)$ with log-prob $\log \pi_\theta(a_t \mid z_t)$    (*diag-Gaussian example:* $\nabla_{a_t} \log \pi = -(a_t - \mu_t) \oslash \sigma_t^2$)
3: **Outer expectation set:** initial observations $\mathcal{O}_0$; **Output:** ESNR
4: **for** $o_0 \in \mathcal{O}_0$ **do**                                                  $\triangleright$ approx. $\mathbb{E}_{o_0}[\cdot]$
5:     $z_0 \leftarrow \text{encode}_\mathcal{M}(o_0)$
6:     **for** $s = 1 \dots S$ **do**                           $\triangleright$ Monte Carlo over action sequences
7:         $z \leftarrow z_0, J \leftarrow 0, d \leftarrow 1$
8:         **for** $t = 1 \dots H$ **do**            $\triangleright$ sample, rollout, and accumulate discounted return
9:            sample $a_t \sim \pi_\theta(\cdot \mid z)$;    $r_t \leftarrow \text{reward}_\mathcal{M}(z, a_t)$
10:           $J \leftarrow J + d \cdot r_t$;    $z \leftarrow \text{next}_\mathcal{M}(z, a_t)$;    $d \leftarrow \gamma \cdot d$
11:         **end for**
12:         **Score gradients via autograd:** for each $t$, compute $\mathbf{s}_t \leftarrow \nabla_{a_t} \log \pi_\theta(a_t \mid z)$
13:         **REINFORCE vector:** $\mathbf{g}^{(s)} \leftarrow \sum_{t=1}^{H} \gamma^{t-1} r_t \, \mathbf{s}_t \ \in \mathbb{R}^{H \cdot A}$
14:     **end for**
15:     $\boldsymbol{\mu} \leftarrow \frac{1}{S} \sum_s \mathbf{g}^{(s)}, \quad \boldsymbol{\sigma}^2 \leftarrow \frac{1}{S-1} \sum_s \big(\mathbf{g}^{(s)} - \boldsymbol{\mu}\big)^2$
16:     $\text{SNR}(o_0) \leftarrow \frac{1}{H \cdot A} \sum_{i=1}^{H \cdot A} \frac{\mu_i^2}{\sigma_i^2 + \varepsilon}$
17: **end for**
18: **return** $\text{ESNR} = \frac{1}{|\mathcal{O}_0|} \sum_{o_0} \text{SNR}(o_0)$

---

---

**Algorithm 2 ESNR** for FoG

---

1: **Inputs:** world model $\mathcal{M}$ (encode, reward, next), horizon $H$, discount $\gamma$, samples $S$
2: **Outer expectation set:** initial observations $\mathcal{O}_0$
3: **Output:** ESNR
4: **for** $o_0 \in \mathcal{O}_0$ **do**                                        $\triangleright$ Approximate $\mathbb{E}_{o_0}[\cdot]$ by averaging
5:     $z \leftarrow \text{encode}_\mathcal{M}(o_0)$
6:     **for** $s = 1 \dots S$ **do**                              $\triangleright$ Monte Carlo over action noise
7:         sample $a_{1:H} \sim \mathcal{N}(0, I)$   (requires_grad)
8:         $J \leftarrow 0, \ d \leftarrow 1, \ z_s \leftarrow z$
9:         **for** $t = 1 \dots H$ **do**                       $\triangleright$ Latent rollout and discounted return
10:           $r_t \leftarrow \text{reward}_\mathcal{M}(z_s, a_t)$
11:           $J \leftarrow J + d \cdot r_t$
12:           $z_s \leftarrow \text{next}_\mathcal{M}(z_s, a_t)$
13:           $d \leftarrow \gamma \cdot d$
14:         **end for**
15:         $\mathbf{g}^{(s)} \leftarrow \nabla_{a_{1:H}} J \in \mathbb{R}^{H \cdot A}$
16:     **end for**
17:     $\boldsymbol{\mu} \leftarrow \frac{1}{S} \sum_{s=1}^{S} \mathbf{g}^{(s)}, \quad \boldsymbol{\sigma}^2 \leftarrow \frac{1}{S-1} \sum_{s=1}^{S} \big(\mathbf{g}^{(s)} - \boldsymbol{\mu}\big)^2$
18:     $\text{SNR}(o_0) \leftarrow \frac{1}{H \cdot A} \sum_{i=1}^{H \cdot A} \frac{\mu_i^2}{\sigma_i^2 + \varepsilon}$
19: **end for**
20: **return** $\text{ESNR} \ = \ \frac{1}{|\mathcal{O}_0|} \sum_{o_0} \text{SNR}(o_0)$

---

---

**Algorithm 3 ESNR (CEM / TD-MPC2)**: SNR of CEM-weighted action "gradients" under a world model

---

1: **Inputs:** world model $\mathcal{M}$ (encode, reward, next), horizon $H$, discount $\gamma$, pop. size $N$, elites $E$, temperature $\tau$, grad samples $S$

2: **Action stats:** mean $\boldsymbol{\mu} \in \mathbb{R}^{H \times A}$ (init. $= \mathbf{0}$), diag std $\boldsymbol{\sigma} \in \mathbb{R}^{H \times A}$ (fixed)

3: **Outer expectation set:** initial observations $\mathcal{O}_0$; **Output:** ESNR

4: **for** $o_0 \in \mathcal{O}_0$ **do**             $\triangleright$ approx. $\mathbb{E}_{o_0}[\cdot]$

5:   **for** $s = 1..S$ **do**         $\triangleright$ Monte Carlo over action populations

6:    sample $\varepsilon \in \mathbb{R}^{N \times H \times A}$ i.i.d. $\mathcal{N}(0,1)$;   $\mathbf{A} \leftarrow \boldsymbol{\mu} + \boldsymbol{\sigma} \odot \varepsilon$

7:    $J_n \leftarrow \sum_{t=1}^{H} \gamma^{t-1} \operatorname{reward}_{\mathcal{M}}(z_t^{(n)}, A_t^{(n)})$ with $z_{t+1}^{(n)} = \operatorname{next}_{\mathcal{M}}(z_t^{(n)}, A_t^{(n)})$, $z_1^{(n)} = \operatorname{encode}_{\mathcal{M}}(o_0)$

8:    **Elites:** keep indices $\mathcal{E}$ of top-$E$ returns; restrict $\{J_n\}$, $\{\mathbf{A}^{(n)}\}$ to $n \in \mathcal{E}$

9:    **Softmax weights:** $w_n \leftarrow \operatorname{softmax}(\tau(J_n - \max_m J_m))$ over $n \in \mathcal{E}$

10:    **CEM action "gradient" (as in code):** $\mathbf{g}^{(s)} \leftarrow \sum_{n \in \mathcal{E}} w_n \mathbf{A}^{(n)} \in \mathbb{R}^{H \times A}$    $\triangleright$ *no $\sigma^{-2}$ factor*

11:   **end for**

12:   stack $\mathbf{g}^{(1:S)} \in \mathbb{R}^{S \times H \times A}$; reshape to $\mathbb{R}^{S \times (HA)}$

13:   $\boldsymbol{\mu}_g \leftarrow \frac{1}{S} \sum_s \mathbf{g}^{(s)}, \quad \boldsymbol{\sigma}_g^2 \leftarrow \frac{1}{S-1} \sum_s (\mathbf{g}^{(s)} - \boldsymbol{\mu}_g)^2$

14:   $\operatorname{SNR}(o_0) \leftarrow \frac{1}{HA} \sum_{i=1}^{HA} \frac{\mu_{g,i}^2}{\sigma_{g,i}^2 + \varepsilon}$

15: **end for**

16: **return** ESNR $= \frac{1}{|\mathcal{O}_0|} \sum_{o_0} \operatorname{SNR}(o_0)$

---

# C   Additional Experimental Results

## Policy Extraction Performance

## Raw Large World Model Planning Results

**Table 3:** Rope task: Minimum Reduced Chamfer Distance (CD) per configuration (1–10) over 10 MPC steps, shown across all epochs.

| Model | Epoch | 1 | 2 | 3 | 4 | 5 | 6 | 7 | 8 | 9 | 10 |
|-------|-------|---|---|---|---|---|---|---|---|---|----|
| | | | | | | Configs (1–10) | | | | | |
| dino | 1 | 0.09459 | 0.04193 | 0.48082 | 0.21772 | 0.11608 | 0.53926 | 0.28854 | 1.03296 | 0.10842 | 0.22533 |
| dino | 25 | 0.11339 | 1.46259 | 0.09250 | 0.99083 | 0.87808 | 0.32004 | 0.23874 | 1.31385 | 0.07392 | 0.26680 |
| dino | 50 | 0.10308 | 0.14979 | 1.05701 | 0.19003 | 0.96528 | 0.13502 | 0.30932 | 0.32733 | 0.07042 | 0.16416 |
| dino | 75 | 0.09642 | 0.21736 | 0.15945 | 0.38226 | 0.77112 | 0.04774 | 0.53940 | 0.19794 | 0.06206 | 0.09256 |
| dino | 100 | 0.07289 | 0.25567 | 0.10827 | 0.15935 | 1.03337 | 0.28235 | 0.35111 | 0.22349 | 0.06045 | 1.13459 |
| dino | 150 | 0.06513 | 0.89921 | 0.65513 | 0.24267 | 0.16757 | 0.11382 | 0.21172 | 0.15394 | 0.09686 | 0.12657 |
| r3m | 1 | 2.41500 | 5.88838 | 2.31113 | 2.84423 | 2.11243 | 3.12261 | 2.27891 | 2.12716 | 1.86786 | 1.38681 |
| r3m | 25 | 0.53223 | 2.17162 | 2.56594 | 1.52169 | 0.94276 | 0.23914 | 1.06509 | 1.95005 | 0.27470 | 1.91671 |
| r3m | 50 | 1.27117 | 1.42981 | 3.24293 | 0.93250 | 2.03880 | 0.70580 | 1.93068 | 1.17572 | 1.62150 | 1.46967 |
| r3m | 75 | 0.68322 | 0.41783 | 1.20674 | 0.56782 | 0.14875 | 0.09701 | 0.42042 | 0.96485 | 0.72479 | 1.62964 |
| r3m | 100 | 1.00865 | 0.21431 | 1.89690 | 0.46082 | 1.33459 | 0.20927 | 0.58501 | 1.52318 | 0.88556 | 2.01774 |
| r3m | 150 | 0.07504 | 2.50585 | 1.12173 | 1.76768 | 0.71625 | 1.32995 | 1.75855 | 1.00132 | 0.85643 | 1.31559 |
| resnet | 1 | 0.35964 | 2.08357 | 1.98353 | 0.66609 | 1.79421 | 1.43631 | 1.53532 | 2.32653 | 0.93529 | 2.65346 |
| resnet | 25 | 1.39610 | 1.92685 | 1.53913 | 1.38198 | 2.69483 | 0.52702 | 1.49306 | 2.02805 | 1.56138 | 1.72422 |
| resnet | 50 | 0.55921 | 3.55125 | 1.48298 | 1.63048 | 1.50783 | 0.39569 | 0.66874 | 0.73747 | 0.81317 | 2.07038 |
| resnet | 75 | 0.49132 | 1.67828 | 3.18766 | 1.50089 | 1.66228 | 0.89216 | 1.65985 | 2.10973 | 0.49975 | 1.99412 |
| resnet | 100 | 1.24596 | 1.59694 | 1.69936 | 1.24382 | 1.67177 | 0.23903 | 0.48933 | 2.19391 | 2.01874 | 2.34193 |
| resnet | 150 | 1.56192 | 1.89500 | 4.26792 | 1.40917 | 1.27476 | 0.38079 | 1.36250 | 4.08078 | 1.69061 | 1.86746 |
| vjepa2 | 1 | 0.09648 | 0.91299 | 1.81233 | 1.35777 | 0.23752 | 0.78538 | 1.44923 | 0.89511 | 0.43192 | 1.59231 |
| vjepa2 | 25 | 0.29945 | 1.83534 | 2.20004 | 1.41878 | 0.16598 | 0.10504 | 1.66310 | 1.52258 | 0.41726 | 1.05544 |
| vjepa2 | 50 | 0.34753 | 1.51121 | 1.78945 | 1.33096 | 0.05454 | 0.64381 | 0.43591 | 1.55015 | 0.34436 | 3.10286 |
| vjepa2 | 75 | 0.19621 | 1.54930 | 1.76645 | 1.38654 | 0.14202 | 0.72667 | 1.08851 | 1.73931 | 0.90187 | 2.27225 |
| vjepa2 | 100 | 0.82727 | 2.84179 | 2.13063 | 1.99864 | 0.11719 | 0.70772 | 1.14918 | 1.71183 | 1.93167 | 2.44238 |
| vjepa2 | 150 | 0.24537 | 2.70079 | 1.88925 | 1.28137 | 0.03945 | 0.40495 | 1.13179 | 1.65183 | 0.20860 | 1.45995 |

**Rope Task Details.** Rope (SoftGym). A 7-DoF arm manipulates a deformable rope to a target configuration. Planning uses MPC with CEM as the inner optimizer. Unless stated otherwise: MPC runs for 10 iterations with prediction horizon 5 and 5 action updates per iteration; CEM draws 100 samples for 10 optimization steps, keeps the top-30 elites, and uses a variance scale of 1.

USING MEAN AS A METRIC

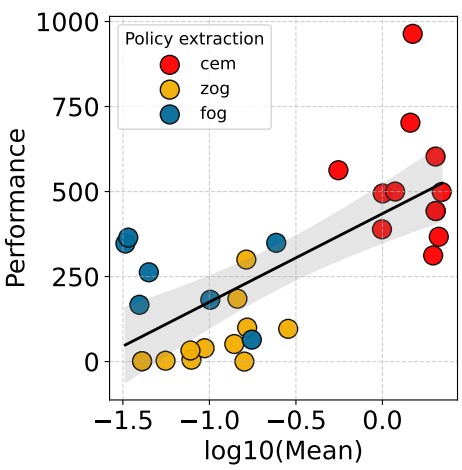 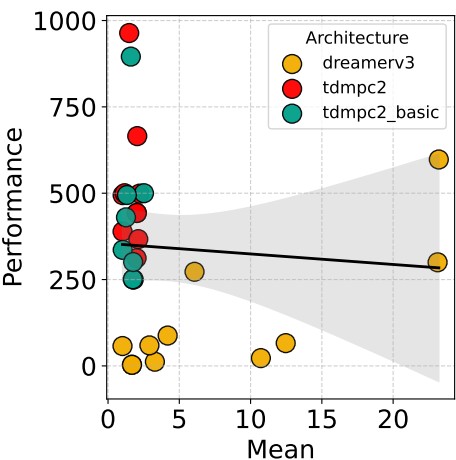

Figure 9: **Mean metric vs. performance.** Relationship between the mean of the policy gradient and downstream policy performance, across policy extraction method (left) and across architectures (right) across all tasks

USING VARIANCE AS A METRIC

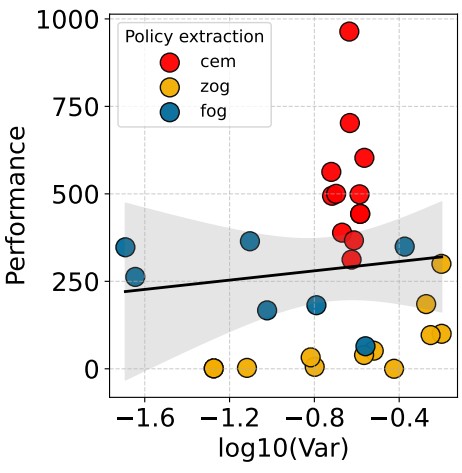 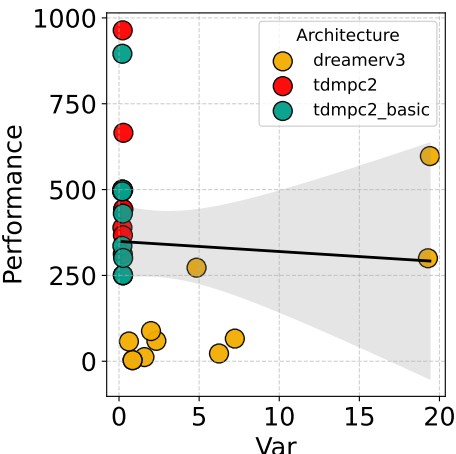

Figure 10: **Variance statistic vs. performance.** Relationship between the variance of the policy gradient and downstream policy performance performance, across policy extraction method (left) and across architectures (right) across all tasks

LARGE WORLD MODEL EVALUATION TASKS AND METRICS

We evaluate our models on three manipulation tasks from the ManiSkill benchmark: PickCube-v1, PushCube-v1, and PokeCube-v1. Each task requires the agent to navigate a 7-DOF end-effector (3D position, 3D orientation, and gripper state) to achieve task-specific goals. For PickCube, the

agent must grasp and lift a cube; for PushCube, the agent must push a cube to a target location; and for PokeCube, the agent must poke a cube with precise contact. We measure performance using the inverse accumulated state difference metric, computed as 1 / (accumulated L2 distance between predicted and goal end-effector states), where the accumulation occurs over all steps in an evaluation episode. This metric rewards models that rapidly converge to goal configurations, with higher values indicating better performance. The state representation consists of the end-effector's 7D pose (xyz position + quaternion orientation + gripper position), and the L2 distance is computed at each timestep and summed across the episode. All results are averaged across 4 environment seeds to ensure statistical robustness.

