# OpenReview forum: "Towards Policy-Aware World Models"
_ICLR.cc/2026/Conference — Submitted to ICLR 2026_

### Official Review · Reviewer_pMJj · 2025-10-29

**Soundness:** 1
**Presentation:** 2
**Contribution:** 1
**Rating:** 2
**Confidence:** 4

**Summary:**

This paper introduces the Expected Signal-to-Noise Ratio (ESNR) as a training-time metric for evaluating world models with respect to downstream policy learning. ESNR measures how well a policy can learn from a world model by quantifying the signal-to-noise ratio of policy gradients. Originally proposed for first-order policy gradients in model-based RL from scratch, ESNR is here extended to arbitrary policy gradient estimators and to pretrained world models. The paper evaluates ESNR across various settings: different policy gradient methods, world model architectures, and pretrained models.

**Strengths:**

- Problem importance: The work targets an increasingly relevant problem: how to assess world models efficiently for MBRL as scaling costs rise. A predictive, policy-aware metric is indeed needed.
- Clarity of the core idea: The main motivation and intuition behind ESNR are straightforward.
- Practical utility: ESNR can be computed without real-environment rollouts or a trained policy, making it computationally efficient and potentially valuable for large world models.
- Empirical breadth: The experiments span multiple policy extraction classes (Zeroth-, First-order gradients, MPC-based methods) and world model types (from DreamerV3 to large pretrained visual encoders), showing effective results.

**Weaknesses:**

1. **Limited novelty over prior ESNR work:**
   The conceptual novelty is modest. Prior works, TPX (Parmas et al., 2023b) and PWM (Georgiev et al., 2025), already proposed and used ESNR to analyze and improve policy gradient estimators in RL from scratch. The core contribution here lies mainly in extending the usage of ESNR to pretrained world models and other policy optimization methods, which is incremental rather than conceptual.

2. **Mathematical clarity and notation issues:**  These disrupt readability and please see my questions below.

3. **Lack of theoretical justification for cross-method comparisons:**
   It remains unclear whether ESNR can validly compare different *policy optimization methods* (e.g., FoG vs. ZoG vs. MPPI), given their qualitatively different gradient behaviors. The “toy example” (Fig. 2) shows FoG and MPPI achieving similar final returns but very different ESNR values, which undermines interpretability. Some theoretical insight into why ESNR should generalize across such gradient estimators will be very helpful.

4. **Missing baselines against model-only metrics:**
   When comparing world models trained with the same policy method, the paper should benchmark ESNR against traditional model quality metrics, e.g., pixel-based (MSE, PSNR) or feature-based (LPIPS), to demonstrate ESNR’s unique predictive power. As shown in Fig. 3, model loss can rankly correlate with policy success rate better than ESNR.

5. **Unclear or contradictory empirical results:**
   - **Biased region and negative trends:** The U-shape ESNR curve and its “biased region” (Fig. 3 and 5) suggest that ESNR may mislead early in training. This undermines its reliability as a general training metric.
   - **Inconsistent intra-method correlation:** In Fig. 4, ESNR correlates across policy classes, but within a single method (e.g., ZOG or DreamerV3) the trend appears weak or even negative. Reporting quantitative correlations (Pearson/Spearman) would clarify these claims.

6. **Lack of discussion on out-of-distribution (OOD) behavior:**
   ESNR estimation could be inflated by gradients on OOD actions from offline dataset, making it unreliable for deployment. The paper does not address this limitation.

**Questions:**

**Mathematical:**
- What is $\mathcal{B}$?
- In Eq. 1, the decoder should output $z_{t+1}$, not $s_{t+1}$; may mark “optional” to decoder for reconstruction-free setups.
- In Eqs. 1–2, clarify whether $s_t$ should be $o_t$; what is $\hat{s}_{t+1}$?
- Eq. 1 only has $\phi$, while Eqs. 2–3 introduce $\psi$, $\theta$.  Please define all parameters consistently.
- Reconstruction-based methods often include reward/value prediction and latent consistency; Eq. 2 should reflect this.
- Eq. 4: should $r$ be $R_\phi$?
- It is better to clarify whether the policy is trained using *only* the world model or a mixture of model/offline/online rollouts in the background section.
- How are initial states sampled? What is the distribution of $z_t$? The planning horizon cannot be infinite.
- Why is Sutton et al. (1999) described as first-order?
- Eq. 6 has nested expectations over $a_t$.
- Eq. 9 overloads $J(\cdot)$ for both policy and action-sequence objectives.
- Eq. 10: since $\nabla_\theta J(\theta)$ is vector-valued, how is ESNR reduced to a scalar?
- Code 1: why does `grad_f` return an additional dimension of $N$? Where does stochasticity arise if dynamics are deterministic?
- L307: should a constant gradient estimator yield ESNR = 0 (due to the small epsilon)?

**Conceptual:**
- Please justify why replacing LayerNorm + ReLU with Mish + SimNorm reduces regularization in modified TD-MPC2.
- Clarify what “performance” refers to—policy success in real environments or within the learned model? Both should be compared to ESNR.

Minor:
- Abbreviations like PVR, PWM, and SNR are not introduced clearly.
- Any citations supporting the claim that first-order gradients have lower variance? If this is true, does this make it unfair to compare FoG with ZoG for ESNR as there is a systematic trend?

---

> ### Author Response · Authors · 2025-11-26
>
> > Limited novelty over prior ESNR work
>
> We thank the reviewer for sharing prior work in this space, particularly the credit assignment line of work which we found very relevant. That being said, those works do not explore world models. Parmas et al. explore using ESNR to compare different policy gradient estimators. We further extend this technique to compare world models and understand when they are sufficiently trained. To the best of our knowledge, this has not been proposed in other works. ESNR has also helped us tremendously in iteratively developing novel vision world models, showing its practical use and importance to development of better methods.
>
> > Mathematical clarity and notation issues: These disrupt readability and please see my questions below.
>
> We thank the reviewer for their extensive feedback! We have corrected and simplified equations in section 3. We address all other concerns below:
>
> > L307: should a constant gradient estimator yield ESNR = 0 (due to the small epsilon)?
>
> A constant gradient estimator will be zero variance, making the denominator 0.
>
> > Eq. 10: since $\nabla_\theta J(\theta)$ is vector-valued, how is ESNR reduced to a scalar?
>
> In the standard ESNR formulation, it is averaged across parameter dimension. We use the action sequence as our "policy parameters", so we average across action trajectory dimension.
>
> > Code 1: why does grad_f return an additional dimension of $N$? Where does stochasticity arise if dynamics are deterministic?
>
> $N$ is the number of action samples; it is stated in the caption of code 1. Stochasticity arises from the randomly initialized / stochastic policy that we sample trajectories under.
>
> > Missing baselines against model-only metrics
>
> We thank the reviewer for their feedback. A key problem identified in world model methods is that lack of regularization (or overfitting) to offline data leads to rough, hard-to-optimize objective landscape. This has been identified in prior works like PWM [1]. In short, prediction loss is misleading if our objective landscape itself is qualitatively bad.
>
> In response to this concern, we have (i) added new experiments in Section 5 (new Fig. 5) that directly compare model loss (and LPIPS) and ESNR against downstream returns, and (ii) clarified this baseline comparison in the abstract and in the introduction of Section 5. Concretely, we first compare TD-MPC2 to a TD-MPC2-basic variant and find that the basic model achieves lower prediction loss but worse converged policy performance, while ESNR correctly identifies TD-MPC2 as the better world model. We then compare DINO-WM against VJEPA2 and find that LPIPS shows weak correlation with downstream performance, whereas ESNR shows strong correlation. We formalize these observations using Spearman correlation coefficients, reported in Figure 5.
>
> >  The U-shape ESNR curve and its “biased region” (Fig. 3 and 5) suggest that ESNR may mislead early in training
>
> The U-shaped ESNR curve is primarily a study on how ESNR evolves during world model training, not a claim that “lower ESNR is always better” throughout training. Our intended usage of ESNR has two stages: (i) along the training cycle of a single world model, ESNR helps identify when the model exits the strongly biased regime (empirically, around the minimum of the U-curve) after which returns begin to increase; and (ii) once we restrict attention to world models that are past this biased region, ESNR can then be used to compare different architectures or training configurations. We have clarified this procedure and its practical implications in Section 5.
>
> >  Inconsistent intra-method correlation
>
> We thank the reviewer for this excellent suggestion. We included Pearson correlation in all experiments where correlation would show meaningful statistical formalism. (Fig 5, Table 2)
>
> In Fig. 4, each point within a method corresponds to a different task, and ESNR is computed as a ratio of signal (mean reward over stochastically sampled actions) to noise (variance of those rewards). Consequently, ESNR is inherently tied to the reward function and scale of each task. This means that absolute ESNR values (and therefore raw Pearson/Spearman correlations) are not directly comparable across tasks, even within the same method: ESNR for DreamerV3 on hopper-hop is not on the same scale as ESNR for cheetah-run. The primary purpose of Fig. 4 is to show that, when we fix a task and vary the world model or policy class, higher ESNR is associated with better policy performance, rather than to claim a strong cross-task correlation within a single method. We will clarify this limitation in the text.
>
> > Lack of discussion on out-of-distribution (OOD) behavior:
>
> We appreciate the reviewer raising this point. We agree that understanding ESNR under OOD actions is interesting. The required experiments to explore this will take longer than what the rebuttal period allows for, but we are committed to adding it in for the camera-ready paper.

---

### Official Review · Reviewer_iBSg · 2025-10-31

**Soundness:** 1
**Presentation:** 2
**Contribution:** 2
**Rating:** 2
**Confidence:** 4

**Summary:**

The paper proposes to understand world model's policy training readiness with expected signal to noise ratio, i.e., the expected ratio of expectation squared and variance of policy gradients. It justifies this metric's usefulness by showing the correlation of ESNR with the average rewards.

**Strengths:**

- The problem this paper tackles is indeed a known problem and a surrogate solution is justified. Evaluating world models with policy training is wasteful and slow. Having a way to know how "trainable" a world model could be helpful.
- I appreciate the toy example mentioned in the paper in Figure 2. It was useful to understand the problem from the authors' perspective.
- The metric is easy to calculate.

**Weaknesses:**

- I am not convinced that the ESNR metric is _actually_ predicting what we want to know from the world model, i.e., whether a trained policy on the world model would be useful. I feel it is still correlated with the accuracy of world model. Would the authors be able to show the clear example where the world model accuracy is fixed (**not** by training epochs) but the loss landscape is different and ESNR still correlates with rewards?

- Equation 2: Reconstruction based world models also learn from minimizing the rewards, e.g., dreamer. Authors include in equation 3 for reconstruction-less world models

- Equation 2 seems sloppy. What is $\hat{s_{t+1}}$? I think author meant to write $\hat{o_{t+1}}$ for reconstructed image? Also, by authors' definition, encoders take $o$, not $s$, but the authors use $E(s_t)$ in the equation.

**Questions:**

See weaknesses.

---

> ### Author Response · Authors · 2025-11-26
>
> > I am not convinced that the ESNR metric is actually predicting what we want to know from the world model
>
> We thank the reviewer for raising this concern. We agree that it is important to show that ESNR is not merely a proxy for world model accuracy. We have added an experiment across three tabletop manipulation tasks that shows the prediction accuracy, illustrated by LPIPS [1], across VJEPA2 and DINOWM model checkpoints are fairly the same (Fig 5.); however, the performance varies across these two models and ESNR mirrors this delta in performance.
>
> We would also like to refer the reviewer to our new “ESNR vs. Pred Loss” experiments (Fig 5) that illustrates prediction accuracy metrics are inversely correlated with performance while ESNR shows good correlation. We tested *these* models across our entire RL test set (14 tasks). This underlines the benefit of having a policy centric metric rather than one that judges the world model representation and loss alone. For all experiments, we also formalize the correlations by presenting the Spearman correlation coefficent.
>
> > Equation 2: Reconstruction based world models also learn from minimizing the rewards, e.g., dreamer. Authors include in equation 3 for reconstruction-less world models
>
> We acknowledge that Dreamerv3 utilizes reward in their world model formulation. And, on the flip side, other reconstruction based methods (like cosmos) don’t utilize reward signals. We appreciate the reviewer pointing this out, and we have updated the equation to reflect that both reconstruction-based and reconstruction-less models can utilize an optional additional signal (eg. reward or Q value) to optimize their world model towards.
>
> > Equation 2 seems sloppy. What is $\hat{s_{t+1}}$? I think author meant to write $\hat{o_{t+1}}$ for reconstructed image? Also, by authors' definition, encoders take $o$, not $s$, but the authors use $E(s_t)$ in the equation.
>
> We thank the reviewer for spotting these inconsistencies. We have updated equation 2 to be consistent in notation and also made it clearer of their meanings.
>
> [1] Zhang, Richard, et al. "The unreasonable effectiveness of deep features as a perceptual metric."

---

### Official Review · Reviewer_VbFX · 2025-11-01

**Soundness:** 1
**Presentation:** 3
**Contribution:** 2
**Rating:** 2
**Confidence:** 5

**Summary:**

This paper explores the question of how to determine whether a world model is good enough to be used for "policy extraction" - learning a policy using the world model through a number of possible model-based reinforcement learning methods. A sure fire way to answer this question is to train a policy with the given model, and evaluate the policy's performance. However, training and evaluating an RL policy end-to-end is prohibitively expensive, especially considering the increasing costs of training large world models on internet-scale data. The paper argues that the expected noise to signal ratio of the policy gradient induced by a world model is a reliable indicator of its effectiveness downstream for policy learning. Given its relatively low compute cost, the author's perform many experiments in simulated robotics environments and note how the ESNR of the policy gradient correlates with downstream policy performance across different world models and policy extraction methods.

**Strengths:**

The paper presents its results and method very clearly, and the structure of the paper is easy to follow. The authors uses a diverse set of world models, environments, and policy extraction methods to make a point, which is necessary due to the experimental nature of the paper, which relies on observations based on correlations. The idea of using the ESNR of a policy gradient induced by a world model is an interesting one that has not been extensively studied to evaluate policy-aware world models, especially when it comes to large pre-trained world models.

**Weaknesses:**

* It is generally well-understood that low gradient variance is imperative to effective downstream learning in machine learning (https://arxiv.org/pdf/2007.04532) and particularly so in reinforcement learning (https://arxiv.org/pdf/2011.09464, https://arxiv.org/pdf/1912.02503) and model-based RL (https://proceedings.mlr.press/v202/parmas23a/parmas23a.pdf) as mentioned in the paper. In fact, the return landscape in RL has also been explored (https://arxiv.org/pdf/2309.14597) as a metric for effective policy updates. The claim that "a general, applicable metric that predicts downstream policy performance across heterogenous world model architectures and policy extraction methods as been lacking" (line 119) is too strong, as the above references show that examining gradient variances (or ESNR as a form of normalized variance) is already an existing metric for evaluation. However, this paper does provide additional empirical evidence for this to be an effective metric for policy-aware world models, it does not suggest a novel metric.
* While the reasoning and intuition behind their method is well motivated, the absence of theoretical evidence requires more robust statistical results. There is no clear evidence why their proposed metric is more effective at predicting downstream performance than existing metrics. See questions for additional details.
* It would be nice to see further comments and discussions around the downstream performance of different world models, and policy extraction methods to better understand why certain methods or models exhibit better or worse ESNRs. For example, in line 364, the authors note that "the gradients [of FoG] will have lower variance and greater expected norm when the objective is well defined". Is this true in the experiments? Why speculate when the data should be readily available? CEM is also noted to have the best performance without further explanations.

**Questions:**

* What makes the ESNR in particular a good indicator of performance? Since the ESNR is the ratio of the scale of the gradient and the variance of the gradient, would either the numerator (scale of policy gradient) or denominator (variance of policy gradient) alone also correlate with the downstream performance? What about the model predictive loss? Figure 3 seems to suggest that the downstream policy success rate is actually also well-correlated with the world model loss. It would be interesting to juxtapose the model loss with downstream performance in all results where possible, since that is how most practitioners evaluate a world model's effectiveness before using it for policy extraction.
* In line 411, the authors note that their experiments "underscore the importance of objective landscapes induced by world models" and that "ESNR mirrors this gap". It would be interesting to expand upon this observation, and perhaps show additional evidence for this mirroring.
* The observation of the U-shape of the ESNR during world model training is interesting, but problematic for the message of this paper. It suggests that the ESNR alone is sufficient to predict downstream performance, since lower ESNR actually results in better performance when in the biased region of the training. Is there a way to know in what region we're in if we only have a single snapshot of a pre-trained world model? The results in Figure 5 are also contradictory to the hypothesis of the paper -- the lowest ESNR actually resulted in the best policies, even in the case of TD-MPC2 which seems to have crossed beyond the boundary of the biased region.

### Minor comments
- Typo in line 95: the generative **capabilities** of [...]
- Confusion in notation in section 3.1: what is $s_t$ and $\hat{s}_t$ are not properly defined.
- Some of the formalism introduced in section 3 is not necessary for the main paper, and introduces unnecessary complexity. The exact mathematical formulations of loses, world model components, different policy gradients and online planning methods is not referenced or discussed again in the results. Consider simplifying and relegating details to the Appendix unless the content is necessary to discuss results.
- The ESNR metric used for experiments, as provided by the pseudo-code does not match the actual ESNR of interest in equation 10. The pseudocode evaluates $\nabla_aJ$, while the actual ESNR of interest should be w.r.t. the policy parameters $\nabla_\theta J$.

---

> ### Author Response · Authors · 2025-11-26
>
> > However, this paper does provide additional empirical evidence for this to be an effective metric for policy-aware world models, it does not suggest a novel metric.
>
> We thank the reviewer for pointing out these related works on credit assignment and return landscapes, which we found very relevant. These papers study gradient variance and related quantities for policy optimization, but do not consider world models or the problem of selecting and early-stopping world models for policy extraction. Parmas et al., for example, use ESNR to compare policy gradient estimators, whereas we apply ESNR to compare world models and identify when they are sufficiently trained for reliable policy extraction, a use case that, to our knowledge, has not been explored. In practice, ESNR has been most useful to us as a model-selection signal when iterating on vision world models, rather than as a new estimator per se. We will revise the statement around line 119 to more clearly reflect this scope and contribution.
>
> > Q: What makes the ESNR in particular a good indicator of performance?
>
> We thank the reviewer for raising this important concern. We added new experiments in Appendix C comparing the mean (numerator) and variance (denominator) metric in isolation against the downstream policy performance. In particular, we notice no strong global trend across policy extraction methods and world model architectures when using mean and variance alone as metrics.
>
> Furthermore, Parmas et al. [1] motivated ESNR by acknowledging that variance alone of the gradient estimator is not suitable for certain algorithms that include bias itself in their gradient estimation (eg: DreamerV3). We have updated our script to restate this fact (line 257).
>
> > What about the model predictive loss?
>
> Conceptually, a key challenge in world model methods is that insufficient regularization or overfitting to offline data can lead to a rough, hard-to-optimize policy objective landscape, even when the predictive loss continues to decrease (as also observed in [1][2][3]). In such cases, prediction loss alone can be a misleading proxy for downstream performance.
>
> In response to this concern, we have (i) added new experiments in Section 5 (new Fig. 5) that directly compare model loss (and LPIPS) and ESNR against downstream returns, and (ii) clarified this baseline comparison in the abstract and in the introduction of Section 5. Concretely, we first compare TD-MPC2 to a TD-MPC2-basic variant and find that the basic model achieves lower prediction loss but worse converged policy performance, while ESNR correctly identifies TD-MPC2 as the better world model. We then compare DINO-WM against VJEPA2 and find that LPIPS shows weak correlation with downstream performance, whereas ESNR shows strong correlation. We formalize these observations using Spearman correlation coefficients, reported in Figure 5.
>
> This illustrates how ESNR provides additional policy-aware signal beyond predictive loss.
>
> > Q: In line 411, the authors note that their experiments "underscore the importance of objective landscapes induced by world models" and that "ESNR mirrors this gap".
>
> We hope our new experiments in Fig 5 helps this.
>
> > Q: The observation of the U-shape of the ESNR during world model training is interesting, but problematic for the message of this paper.
>
> We appreciate the reviewers for raising this point. The U-shaped ESNR curve is primarily a study on how ESNR evolves during world model training, not a claim that “lower ESNR is always better” throughout training. Our intended usage of ESNR has two stages:
>
> (i) Along the training cycle of a single world model, ESNR helps identify when the model exits the strongly biased regime (empirically, around the minimum of the U-curve) after which returns begin to increase
>
> (ii) Once we restrict attention to world models that are past this biased region, ESNR can then be used to compare different architectures or training configurations.
> We have changed our script to clarify this procedure and its practical implications in Section 5.
>
> > Q: “Is there a way to know in what region we're in if we only have a single snapshot of a pre-trained world model?”
>
> Just as it is difficult to diagnose underfitting vs. overfitting from a single value of prediction loss, it is similarly hard to localize a model on the ESNR U-curve without some notion of training trajectory or multiple checkpoints. In this work we therefore focus on using ESNR across checkpoints to identify the unbiased region and to compare models within that region. We acknowledge that developing diagnostics that reliably infer this regime from a single snapshot is an interesting direction for future work.
>
> [1] Paavo Parmas et al. Proceedings of the 40th International Conference on Machine Learning
> [2] Cui, Brandon et al. "Control-aware representations for model-based reinforcement learning.".
> [3] Hansen, Nicklas et al. "Td-mpc2: Scalable, robust world models for continuous control."

---

### Official Review · Reviewer_HHnq · 2025-11-01

**Soundness:** 2
**Presentation:** 2
**Contribution:** 2
**Rating:** 4
**Confidence:** 3

**Summary:**

This paper introduces the Expected Signal-to-Noise Ratio (ESNR) of policy gradients as a novel, efficient, training-time metric for predicting the downstream policy performance of world models in robotics. The central problem addressed is the current costly and slow iterative process of training a world model and then fully evaluating its quality via policy learning and environment rollouts, which can take days or weeks. The authors define the concept of Policy Awareness as how well a policy can learn from a model, and ESNR is proposed as an on-the-fly, policy-agnostic proxy for this metric.

**Strengths:**

1. ESNR can be computed with minimal overhead without a trained policy or environment rollouts.

2. The manuscript provides a much-needed policy-centric lens for evaluating world models, which goes beyond standard prediction quality metrics.

**Weaknesses:**

1. This is an incremental work, as shown in the manuscript, the ESNR metric has been proposed by [1].

2. The authors state that they have not yet evaluated ESNR extensively on large-scale world model tasks with billions of parameters. Since the advantage claimed in the paper is speed, experiments should be conducted on larger world models, and we cannot be certain that small-scale conclusions can be extended to the world models of larger world models.

[1] Paavo Parmas, Takuma Seno, and Yuma Aoki. Model-based reinforcement learning with scalable composite policy gradient estimators. In International Conference on Machine Learning, pp. 27346–27377. PMLR, 2023a.

**Questions:**

Is there a model-agnostic or task-agnostic heuristic that can be used to computationally identify the transition point out of the "biased region" (i.e., the ESNR minimum) without needing to wait for a full policy evaluation at a later checkpoint? Since ESNR is meant to save evaluation time, relying on the full U-shaped curve defeats the purpose.

---

> ### Author Response · Authors · 2025-11-26
>
> We thank the reviewer for their valuable feedback. We address your comments in the following.
>
> > The authors state that they have not yet evaluated ESNR extensively on large-scale world model tasks with billions of parameters. Since the advantage claimed in the paper is speed, experiments should be conducted on larger world models, and we cannot be certain that small-scale conclusions can be extended to the world models of larger world models.
>
> We understand the reviewer’s concern about scaling to larger models. In the current version of the paper, we already evaluate ESNR on large pretrained vision-based world models, including DINO-WM (\~ 304M params) and V-JEPA2 (\~1.2B parameters, and show that ESNR remains predictive of downstream performance in this setting (Section 5, new Figure 7). In the revision, we have added additional tasks in the large world model setting to further strengthen this claim (Section 5, Figure 5).
>
> > Q: Is there a model-agnostic or task-agnostic heuristic that can be used to computationally identify the transition point out of the "biased region" (i.e., the ESNR minimum) without needing to wait for a full policy evaluation at a later checkpoint? Since ESNR is meant to save evaluation time, relying on the full U-shaped curve defeats the purpose.
>
> We thank the reviewer for this helpful observation. Our empirical results in Figures 6 show that the ESNR curve exhibits a consistent U-shape across architectures and tasks, with the minimum aligning with the point at which the model exits the biased region and evaluation rewards begin to increase. This means that, in practice, one does not need to wait for the full U-shaped curve or a late policy evaluation: policy extraction can be triggered once ESNR reaches its minimum and starts to rise again. We have revised Section 5 to make this procedure and its practical implications more explicit (new line 477).

---

### Author Response · Authors · 2025-11-26
**General Response**

We thank the reviewers for their insightful feedback. We appreciate that the reviewers identified several key strengths in our work:

- Presented our core idea clearly (reviewers VbFX, pMJj)
- Targeted a well-known and increasingly relevant problem (reviewers HHnq, VbFX, pMJj)
- Rigorous in its experiments and effective in its results (reviewers VbFX, pMJj)
- Presents an efficient, easy-to-calculate metric (reviewers HHnq, iBSg, pMJj)

Despite this positive outlook, a common concern was the lack of baseline comparisons: prediction loss and representation loss (LPIPS) are the de facto policy-centric metrics used in current practice.

We have addressed the reviewers’ concerns individually and updated our manuscript with the below changes (we also include a version comparison PDF in the supplementary material)

**Summary of revisions**:

- **Metric baseline comparison**: We now compare prediction loss and ESNR across two RL world models, and compare LPIPS and ESNR across two larger world models with different representation spaces. In both cases, ESNR succeeds where the baseline metrics fail (new Fig. 5), which addresses the common concern of reviewers [VbFX](https://openreview.net/forum?id=Ro2eG1RRde&noteId=qS28XKAYXS), [iBSg](https://openreview.net/forum?id=Ro2eG1RRde&noteId=V284SW16BK), and [pMJj](https://openreview.net/forum?id=Ro2eG1RRde&noteId=XEsz0wzrN4)

|Metric | Spearman Correlation with Performance |
|---|---|
| ESNR | 0.571 |
| Prediction Loss | -0.520 |

|Metric | Spearman Correlation with Performance |
|---|---|
| ESNR | 1.00 |
| LPIPS | -0.333 |

- **New tasks for larger world models**: Because larger world models often require extensive compute time to evaluate and are a more valuable use-case for ESNR as a metric, we add three new tasks from the ManiSkill test suite (new Fig. 5) to increase our result credibility as suggested by reviewers [HHnq](https://openreview.net/forum?id=Ro2eG1RRde&noteId=a7WW91gn0g) and [VbFX](https://openreview.net/forum?id=Ro2eG1RRde&noteId=qS28XKAYXS)
- **Statistical correlation to verify claims**: Across all relevant experiments, we add statistical correlation analyses to verify our claims rigorously (Table 2 and new Fig.5) as suggested by reviewer [VbFX](https://openreview.net/forum?id=Ro2eG1RRde&noteId=qS28XKAYXS).
- **Policy gradient mean and variance as metrics**: We empirically test using the policy-gradient mean (numerator of ESNR) and variance (denominator of ESNR) as standalone metrics as suggested by reviewer [VbFX](https://openreview.net/forum?id=Ro2eG1RRde&noteId=qS28XKAYXS). Their lack of correlation with downstream performance underlines the importance of ESNR’s formulation (Appendix C addition).
- Rewording and improved clarity in section 3

We believe these revisions strengthen our claim that ESNR provides an effective policy-centric metric that succeeds where other metrics fail.

---

### Author Response · Authors · 2025-11-29
**Note to AC**

We thank the reviewers for their suggestions. Many of the comments focused on the lack of comparisons with existing metrics and on the need for stronger statistical evidence.

In response, we have added new experiments and now report formal correlation metrics across all relevant settings. We believe these revisions substantially strengthen the paper and are grateful to the reviewers for helping improve our work.

---

### Meta-Review · Area_Chair_5Vuo · 2025-12-24

**Summary:**

This paper proposes ESNR as a training-time metric for evaluating the perception ability of World Models strategies, aiming to address the issue that traditional prediction losses are not correlated with downstream policy performance, which leads to an overly long model evaluation cycle. Through experiments, it demonstrates that ESNR effectively predicts the learning effect of policies and guides early stopping in the training of world models. However, both reviewers pMJj and VbFX pointed out the lack of innovation in this paper. Meanwhile, reviewer VbFX raised doubts about the reliability of the U-shaped curve of the ESNR metric. Although the authors completed the benchmark tests and proved that ESNR remains effective when prediction losses fail, and expanded the model scale, I still think their responses to the core issues mentioned above are insufficient.  I encourage the authors to further address these two problems and resubmit. Thus, I recommend rejection.

**Reviewer Concerns:**

Resolved Reviewer Comments:

1.Insufficient experimental benchmarks: The author supplemented the key experiments (Figure 5) in the reply, demonstrating that ESNR can still maintain a high Spearman correlation even when prediction loss and LPIPS fail.

2.Large model scalability: The authors added tests on the 1.2B parameter VJEPA2 model and ManiSkill tasks, demonstrating the effectiveness of this metric in large-scale visual representations and its computational speed advantage.

Unresolved Reviewer Comments:

1.Innovative issue: Although the application scenarios are different, the basic mathematical framework of ESNR has already been presented in Parmas et al. (2023). The reviewers believe that the methodological leap from "contrastive gradient estimators" to "evaluating world models" is not significant enough and falls into the category of incremental innovation.

2.The U-shaped curve issue of ESNR: Although the authors explained that the "minimum value" serves as an early stopping signal, the artificially high values of ESNR in the early training stage (bias region) still raised doubts among the reviewers regarding its reliability as a universal metric.

**Reviewer Scores:**

Since the reviewer did not participate in the subsequent discussions or provide further feedback after the author's submission of the response, based solely on the quality and shortcomings of the author's response, the estimated score for this paper should be between 4 and 5. For specific reasons, refer to Reviewer Concerns.

---

### Decision · Program_Chairs · 2026-01-26

Reject